# Effect of Fermentation Scale on Microbiota Dynamics and Metabolic Functions for Indigo Reduction

**DOI:** 10.3390/ijms241914696

**Published:** 2023-09-28

**Authors:** Nowshin Farjana, Hiromitsu Furukawa, Hisako Sumi, Isao Yumoto

**Affiliations:** 1Bioproduction Research Institute, National Institute of Advanced Industrial Science and Technology (AIST), Sapporo 062-8517, Japan; nowshin.farjana2404@gmail.com; 2Laboratory of Environmental Microbiology, Graduate School of Agriculture, Hokkaido University, Sapporo 060-8589, Japan; 3Sensing System Research Center, National Institute of Advanced Industrial Science and Technology (AIST), Tsukuba 305-8565, Japan; h-furukawa@aist.go.jp; 4North-Indigo Textile Arts Studio, Otaru 047-0022, Japan; hisako.sumi@icloud.com

**Keywords:** indigo reduction, alkaliphile, convergence of microbiota, extracellular electron transport

## Abstract

During indigo dyeing fermentation, indigo reduction for the solubilization of indigo particles occurs through the action of microbiota under anaerobic alkaline conditions. The original microbiota in the raw material (*sukumo*: composted indigo plant) should be appropriately converged toward the extracellular electron transfer (EET)-occurring microbiota by adjusting environmental factors for indigo reduction. The convergence mechanisms of microbiota, microbial physiological basis for indigo reduction, and microbiota led by different velocities in the decrease in redox potential (ORP) at different fermentation scales were analyzed. A rapid ORP decrease was realized in the big batch, excluding *Actinomycetota* effectively and dominating *Alkalibacterium*, which largely contributed to the effective indigo reduction. Functional analyses of the microbiota related to strong indigo reduction on approximately day 30 indicated that the carbohydrate metabolism, prokaryotic defense system, and gene regulatory functions are important. Because the major constituent in the big batch was *Alkalibacterium pelagium*, we attempted to identify genes related to EET in its genome. Each set of genes for flavin adenine dinucleotide (FAD) transportation to modify the flavin mononucleotide (FMN)-associated family, electron transfer from NADH to the FMN-associated family, and demethylmenaquinone (DMK) synthesis were identified in the genome sequence. The correlation between indigo intensity reduction and metabolic functions suggests that V/A-type H^+^/Na^+^-transporting ATPase and NAD(P)H-producing enzymes drive membrane transportations and energization in the EET system, respectively.

## 1. Introduction

There are two main types of dyeing raw materials with indigo that are primarily determined by the fermentation process for the solubilization of indigo: one using indigo extracted from plants and the other using composted indigo-containing plants. The traditional Japanese procedure uses the composted indigo plant *Polygonum tinctorium* L. (*sukumo*) [1,2,3]. Another composted indigo-containing plant, coached woad, which is composted indigo-containing woad (*Isatis tinctoria* L.), was developed in Medieval Europe [4,5,6,7,8]. Composted plants are convenient for long-term storage and transportation. Additionally, they contain seed microorganisms and substrates for indigo reduction fermentation [1,9]. The latter contains plant-decomposed residues and dead bacterial cells generated during production. However, reducing indigo during fermentation using composted indigo-containing plants is sometimes difficult. This suggests that the original microbiota in *sukumo* changes depending on the preparation and maintenance procedures [10]. Although 0.048% operational taxonomic units (OUTs) belonging to the indigo-reducing genus *Alkalibacterium* were previously detected in *sukumo* [9], *Alkalibacterium* OTUs were not detected in >30,000 reads of 16S rRNA gene sequences in next-generation sequencing (NGS). In addition, the microbiota in *sukumo* differs depending on the year of production and the production procedures of the craftsperson. An understanding of environmental factors for the convergence of the microbial community in *sukumo* to induce indigo reduction is important for appropriately controlling the microbiota regardless of the raw material. In indigo-reducing fermentation, *sukumo* is treated with wood ash extract (≥pH 11), following which the aerobic bacteria consume oxygen in the fermentation fluid using substrates present in *sukumo* [9]. This initial change in fermentation activates microbiota adaptable to alkaline and low-ORP environments (≤−600 mV) in the fermentation fluid. In our previous study, high pH, low redox potential (ORP), and the addition of a substrate (wheat bran) induced microbiota to direct the original microbiota in *sukumo* to reduce indigo [10]. However, these parameters change depending on the fermentation procedures, such as the fermentation scale and timing of wheat bran addition.

In most industrial microorganism utilization, batch fermentation of a single strain is carried out in a short period. In contrast, high sustainability and resiliency are observed in the performances of microbial community applications [11,12,13]. However, the microbiota changes depend on ambient environmental changes. Therefore, studies on the transitional changes in the microbiota depending on the fermentation phase are important for understanding and regulating microbial ecosystems for utilization. Small-scale batch experiments have shown that indigo-reducing bacterial communities are constructed within five days by replacing the initially appearing aerobic microbes. In addition, the changing velocity of the microbiota is faster until approximately day 30, regardless of the timing of wheat bran introduction [10,13]. This is probably because the influence of the substrates derived from *sukumo* was effective for approximately 30 days. After approximately day 30, the slowly circulating ecosystem originating from slowly consuming substrates containing wheat bran and residues derived from *sukumo* sustains indigo reduction [13].

Although understanding the dynamics and ecosystem of the microbiota is directly related to the convergence and sustainability of this indigo-reducing microbiota, understanding the characteristics of the constituent microorganisms is also important for understanding the functions of the microbiota in the ecosystem. Transitional changes in microbiota in indigo fermentation employing *sukumo* have been analyzed using PCR-DGGE and clone libraries analysis [2] and NGS in small-scale fermentation in the laboratory [9,10,11,12,13,14]. In addition to these uncultured procedures, indigo-reducing bacteria have been isolated [1,15]. The reduction in indigo was not always observed in the presence of indigo-reducing species. It is thought that the reason for this is as follows: (1) The existing ratio of indigo-reducing bacteria was not sufficiently high. (2) Not ready to metabolize available substrates in indigo-reducing bacteria to produce extracellular electron transfer (EET). (3) The prerequisites for metabolic features (e.g., fulfillment for sufficient concentration of electron mediators) of EET have not been accomplished yet. (4) The background ORP of the fermentation solution was not sufficiently low for the occurrence of EET. From days 2 to 5, the ORP decreased to less than −550 mV, the existing ratio of indigo-reducing species increased, and indigo reduction was achieved. In the early phase of the indigo-reducing state, facultative anaerobic or aerotolerant species, such as *Amphibacillus* spp., *Alkalihalobacillus* spp., and *Alkalibacterium* spp. appear. In addition, obligate anaerobes such as *Alkalicella* spp., *Tissiellaceae,* and *Alkaliphilus* spp. subsequently appear. However, after approximately one month, the ratio of obligate anaerobes tends to decrease, and wheat bran utilizable taxa such as *Amphibacillus* spp. or *Polygonibacillus* spp. tend to predominate [12,13].

Although many indigo-reducing bacteria have been isolated [1], the indigo reduction at the molecular level has not yet been fully understood. It can be predicted that the electron transfer system originating from NAD(P)H reduces indigo because this molecule is the most abundant electron source in the cells. However, as NAD(P)H is required for intracellular metabolisms, it is difficult to explain the transfer of intracellular electron donors to extracellular electron accepters via an enzyme accepting electrons from intracellular NAD(P)H. Nicholson and John reported that indigo particle size decreased when the particles were incubated with the culture supernatant from *Clostridium isatidis* [14]. In addition, Nakagawa et al. found that anthraquinone (AQ) is an effective mediator of indigo reduction activity in various indigo-reducing bacterial strains [15]. These results suggested that indigo reduction occurred via an extracellular mediator in the fermentation fluid. A flavin-based EET system has been reported in *Listeria monocytogenes* [16]. Electrons originating from NADH are transferred to FMN in the flavoprotein associated with the outer surface membrane via the DMK pool in the membrane. Extracellular electron acceptors accept electrons via a free molecular flavin shuttle that is reduced by flavoproteins. Eight genes related to EET are serially coded in the genome of *L*. *monocytogenes*. Corresponding serial gene sequences exist in many taxa belonging to the phylum *Bacillota* (formerly *Firmicutes*). Because bacteria belonging to *Bacillota* are major members of the microbiota in the indigo fermentation fluid, it is possible that indigo-reducing bacteria possess the EET system found in *L*. *monocytogenes*.

In this study, differences in the fermentation scale were examined as environmental factors to converge the microbiota in indigo fermentation. Herein, we show that the rapid ORP decrease can be attributed to the big fermentation scale convergence of indigo-reducing microbiota. Functional analyses of the microbiota related to indigo reduction indicated that carbohydrate metabolism, prokaryotic defense system, and gene regulatory functions are important. These results indicated that bacterial functions such as the metabolic capacity of carbohydrates, gene regulatory functions, and prokaryotic defense system are associated with environmental changes, including low ORP, during the convergence of the microbiota. We show that these events cause indigo reduction. A gene series for EET found in *L*. *monocytogenes* was also found in *Alkalibacterium pelagium*, which was the predominant member in the big batch fermentation. It is believed that the metabolic function of the Kyoto Encyclopedia of Genes and Genomes (KEGG) orthology, which is highly correlated with dyeing intensity, facilitates indigo reduction. This suggests that indigo reduction is highly correlated with the metabolic turnover of indigo-reducing bacteria.

## 2. Results

### 2.1. Changes in Microbial Community in Big and Small Batches

Duplicate samples were obtained from four different fermentation periods (days 3–212) in a big batch. From these eight samples, 434,197 raw sequences (average: 54,275 sequences/sample) were obtained. After trimming the adapters and primers, merging, and quality checks, 163,562 sequences (an average of 20,445 sequences/sample) were obtained.

On day 3 (48 h after fermentation initiation), indigo reduction was observed (dyeing intensity (*L***a***b** value × 10): 1.03) in the big batch, and the ORP was −640 mV (Figure 1, Appendix A and Appendix A). The most predominant taxon was *Alkalibacterium* (38.0% in the ratio). On day 10, indigo reduction was enhanced (dyeing intensity: 1.94), and the ratio of *Alkalibacterium* also increased (45.5%). In addition, *Alkalicella* slightly increased (8.8 → 10.1%), and *Tissierella* increased (2.2 → 7.0%). On day 27, the indigo reduction further increased (dyeing intensity: 2.41). Although the ratio of *Alkalibacterium* slightly decreased (41.3%), *Amphibacillus* increased (2.0 → 7.0%). On day 212, the dyeing intensity (0.51) decreased, and the ORP (−562 mV) increased. *Alkalibacterium* largely decreased (16.7%), and *Erysipelothrix* (0.2 → 8.7%) and *Halomonas* (0 → 9.2%) increased from day 27.

Duplicate samples (except for day 5) were obtained from eight different fermentation periods (days 2–200) in the small-scale batch. From these 15 samples, 768,419 raw sequences (average: 51,228 sequences/sample) were obtained. After trimming the adapters and primers, merging, and performing quality checks, 218,876 sequences (average: 14,592 sequences/sample) were obtained.

On day 2, both the dyeing intensity (0.57) and ORP (−353 mV) were not sufficient for use in the small-scale batch (Figure 1, Appendix A and Appendix A). The predominant taxon was *Bacillus* (13.6%), followed by *Stackebrandtia* (phylum *Actinomycetota*; 6.6%). On day 5, both the dyeing intensity (0.90) and ORP (−598 mV) changed to a favorable value. Taxa belonging to the phyla *Actinomycetota* and *Pseudomonadota* and genus *Bacillus* were decreased. In contrast, *Alkalicella* (0 → 33.4%), *Alkalibacterium* (0 → 9.2%), and *Enterococcus* (0 → 7.8%) dramatically increased. The constituent taxa were similar on days 5 and 7. The ratio of *Alkalicella* increased on day 7 (49.9%). The dyeing intensity increased toward day 7 (1.85), and the ORP decreased to −618 mV. From days 7 to 29, although the ORP increased slightly (−608 mV), the dyeing intensity increased (2.25). *Alkalicella* (18.5%) and *Enterococcus* (1.5%) decreased, whereas *Alkalibacterium* (30.2%) and *Amphibacillus* increased (13.5%). On day 96, the dyeing intensity (1.57) and ORP (−550 mV) changed in unfavorable directions. The ratios of *Alkalicella*, *Alkalibacterium*, *Enterococcus*, and *Amphibacillus* decreased, whereas those in the phylum *Pseudomonadota*, such as *Alcaigenes* and *Burkholderiaceae*, increased. On day 200, the dyeing intensity decreased slightly (1.49), and the ORP increased slightly (−561 mV).

### 2.2. Comparison of Microbiota between Big- and Small-Scale Fermentation Batches in Early Fermentation Phase

The differences in the microbiota within the early fermentation phase are considered to be responsible for the differences at the fermentation scale. The initial transitional changes in the microbiota occurred faster in the big batch than in the small batch. Therefore, days 2–27 in the big batch were compared with days 6–29 in the small batch using the linear discrimination analysis effect size (LEfSe) (Figure 2). *Alkalibacterium* (*Carnobacteriaceae*) was selected in the big batch. In contrast, three taxa involved in the order *Actinomycetes* were predominant in the small batch. In addition, the genera *Alkalicella* and *Enterococcus* and order *Thermomicrobiales* are predominated in the small batch.

### 2.3. Changes in Alpha Diversities

Changes in alpha diversity at sample depths of 5709 and 5708 for big and small batches, respectively, based on the observed operational taxonomic units (OTUs) and Shannon index, are shown in Appendix A. Similar changes were observed in the big and small batches. The range of observed OTUs and Shannon index for the big batch decreased from day 3 to day 10; thereafter, there was a slight increase in observed OTUs to day 27, whereas a change was scarcely observed in the Shannon index. Finally, the observed OTUs decreased, whereas the Shannon index increased from days 27 to 212.

More intense fluctuations in both the observed OTUs and Shannon index were observed in the small-scale batch than in the big-scale batch. Decreasing tendencies from days 2 to 7 and increasing tendencies from days 7 to 14 were noted in the observed OTUs and Shannon index. Although changes were scarcely noted in the observed OTUs during days 14 to 29 and days 96 to 200, decreases in the Shannon index were noted. Both observed OTUs and the Shannon index increased between days 29 and 96.

### 2.4. Relationship between Environmental Variables and Microbial Communities Contributing to Dyeing Intensity in Big- and Small-Scale Batches

The relationship between the environmental parameters and microbial communities contributing to indigo reduction was estimated using redundancy analysis (RDA; Figure 3). Comparing the microbiota of big- and small-scale batches, the microbiota in the big-scale batch on day 3 had already converged to the indigo-reducing state, which was attributed to the rapid change to negative ORP (−640 mV). On the other hand, the slow reduction in ORP in the small-scale batch allowed the adaptation of several taxa, mainly *Actinomycetota*, adaptation in low ORP. Therefore, the convergence of the microbiota toward the indigo-reducing state required more time and steps in the small-scale batch. The changing range of the microbiota in the big batch was smaller than that in the small batch. This was attributed to the rapid change to low ORP at the beginning of the fermentation. The taxa located in the same directions as the dyeing intensity were thought to contribute to indigo reduction. These taxa include *Alkalibacterium*, *Alkalicella*, *Amphibacillus*, *Tissierella*, and *Enterococcus*. Because a big batch rapidly creates a low-ORP environment, it is easy to converge *Alkalibacterium* and *Alkalicella*. The direction of the low ORP was the same as that of the dyeing intensity. The velocity of the transitional change in the big batch was slower than that in the small batch throughout the fermentation period.

### 2.5. Differences in Microbial Interaction Network between Big- and Small-Scale Fermentations

Microbial interactions in the big- (Figure 4A) and small-scale (Figure 4B) fermentation batches were estimated. Microbial interactions consisting of 13 taxa, including a wide range of taxon categories (obligate anaerobic *Bacillota*, lactic acid bacteria, *Bacillales*, other *Bacillota*, *Actinomyceta*, and *Pseudomonodota*), were constructed because of the rapid decrease in ORP in the big batch. In a later phase, a smaller network consisting of eight taxa was also constructed. This network mainly comprised obligate anaerobic *Bacillota* (*Caldicoprobacter*, *Tissierella*, *Fastidiosipila*, *Eubacteriales*). No negative correlation was observed in the big fermentation batch.

In contrast, the network consisted of 20 taxa, mostly consisting of *Bacillales*, other *Bacillota*, *Actinomyceta*, and *Pseudomonodota* transferred to other networks. It is thought that a network consisting of 20 taxa (Figure 4B, right) can metabolize oxygen. *Amphibacillus* and *Tissierella* were negatively correlated with *Azoarcus* (*Pseudomonadota*). In addition, *Amphibacillus* exhibited a negative correlation with *Glycomyces*, *Streptomyces*, and *Stackebrandtia* (*Actinomycetes*), *Halomonas* (*Pseudomonadot*) and *Thermomicrobiales* (*Thermomicrobiota*). The network constructed during the later fermentation phase, which consisted of 13 taxa (Figure 4B, left), was negatively correlated with the network constructed during the early phase. Indigo-reducing taxa that appeared in the later phase, such as *Alkalicella* and *Enterococcus*, exhibited negative correlations with *Pussillimonas* (*Pseudomonadota*) and *Caldicoprobacter* (obligate anaerobic *Bacillota*) and *Pussillimonas*, respectively, in the large network. The alpha diversity (Shannon index; Appendix A) in the small batch tended to increase, and the size of the network decreased. On the other hand, the alpha diversity (Shannon index; Appendix A) tended to decrease in the big batch, and the size of the network also decreased.

### 2.6. Functional Abundance for Requiring Strength of Indigo Reduction

Because the dyeing intensity became weak on day 212 (dyeing intensity: 0.51; Figure 1) and days 10 (dyeing intensity: 1.94) and 27 (dyeing intensity: 2.41) exhibited high dyeing intensity, the ratios of subpathway percentage exhibited ≥ 1.05 on either day 10/day 212 (D10/D212) or day 27/day 212 (D27/D212) in the big batch were selected (Appendix A). In addition, to understand the necessity of enhancing the dyeing intensity within the first month, day 10/day 3 (dyeing intensity: 1.03) (D10/D3) and day 27/day 3 (D27/D3) were estimated. The D10/D212 and D27/D212 ratios were much higher than day 10/day 3 (D10/D3) and day 27/day 3 (D27/D3). The listed subpathway in Appendix A is important for strong dyeing intensity, especially “prokaryotic defense system”, “starch and sucrose metabolism”, and “replication, recombination, and repair proteins” exhibited that the average ratios in D10/D212 and D27/D212 are higher than 1.4. The contributions of the three subpathways of the major bacteria on day 27 were estimated (Table 1). The predominant *Alkalibacterium* (41.3%) exhibited the highest contribution to three subpathways, and the functional contribution ratios were equally high (>64%) in the microbiota. Although the functions of the subpathway of “starch and sucrose metabolism” (14.0%) were higher than the existing ratio in *Amphibacillus* (12.2%), the functional contribution ratios in the microbiota for “prokaryotic defense system” (9.1%) and “replication, recombination, and repair proteins” (8.3%) were lower than the existing ratio. The functional contribution ratios of obligate anaerobes *Tissierella* and *Alcalicella* to “starch and sucrose metabolism” were much lower in each existing ratio in the microbiota.

The subpathways exhibiting the ≥1.05 ratio in either D7 (dyeing intensity: 1.85)/D2 (dyeing intensity: 0.57) or D29 (dyeing intensity: 2.25)/D2 were selected in the small batch (Appendix A). The selected subpathways were similar to those in the big batch. In addition to the subpathways of “prokaryotic defense system”, “starch and sucrose metabolism”, and “replication, recombination, and repair proteins”, which exhibited a high ratio in D10/D212 and D27/D212, “phosphotransferase system (PTS)” was the most important item for indigo reduction (the average ratio in D7/D2 and D29/D2 was 2.44). In comparison between the highest staining date (day 29) and aged fermentation (day 96 or day 212) (Appendix A), high ratios in D29/D96 or D29/D200 were found in “prokaryotic defense system”, “PTS”, “starch and sucrose metabolism” and “cell growth”. The contributions of the six subpathways (in addition to the five subpathways, “replication, recombination, and repair proteins” was considered) of the major constituted bacteria on day 29 were estimated (Appendix A). The predominant *Alkalibacterium* (30.2%) contributed the most to the selected subpathways, except for “cell growth”. Considering constituted ratios of obligate anaerobes *Alkalicella* and *Tissierella*, the functional contribution for “PTS”, “starch and sucrose metabolism”, and carbohydrate metabolism was low.

### 2.7. Identification of Genes Related to EET

As the OTUs exhibiting 100% gene sequence identities (427/427) with *Alkalibacterium pelagium* were 83.0–91.0% in the genus *Alkalibacterium* (39.4–47.2% in the microbiota) identified in the big batch during days 10–27 and the whole genome sequence has already been deposited in the National Center for Biotechnology Information (NCBI) (accession number: NZ_FNZU00000000), we attempted identification of the gene sequence locations related to EET found in *Listeria monocytogenes* [16] in the genome sequence of *Alkalibacterium pelagium*. The gene sequences for the protein translocation of FAD across the membrane and modification of PplA (*ecfA* and *A’*, *fmnA*, and *apbE*/*fmnB*) were found at relatively similar locations in the genome sequence (Figure 5). Several gene sequences of proteins associated with the electron transfer from intracellular NADH to the extracellular FMNylated protein, PplA (*eetA*, *eetB*, and *ndh-2-*like), were also found in similar locations in the genome. However, the gene encoding extracellular FMNylated protein (PplA) was not found. This protein also was also absent in *Aerococcus christensebii* [17] (Figure 5). The genes encoding the production of demethlmenaquinone (DMK), which is membrane-associated and contributes to the separated electron transfer from aerobic respiration, *dmkA* and *dmkB* were found at locations far from other EET genes.

The OTUs exhibiting 99.0–100% identities with *Alkalicella caledoniensis* consisted of 47.2% identity in the genus *Alkalicella* (49.5–50.4% identities in the microbiota) identified in Figure 1 in the small batch at day 7. We also attempted to identify the gene sequence locations related to EET in previously deposited corresponding gene sequences (accession number: NZ_CP058559.1). Some of the genes for FAD transfer components (*ecfA* and *A’*, *fmnA*, and *apbE*/*fmnB*) and electron transfer components (*eetB* and *eetA*) were found (Figure 5). In addition, molybdenum cofactor sulfurase C-terminal (MOSC) domain-containing protein and NADH dehydrogenase, which is not similar to ndh-2, were located near *eetB*, *eetA*, and *apbE*/*fmn*. However, the gene-coding regions of *eetB*, *eetA*, and *apbE*/*fmnB* were far separated from those of *fmnA* and *ecfA* and *A’* and *pplA* were not detected. In addition, processing genes for DMK, *dmkA* and *dmkB*, could not be identified.

Although the genus *Enterococcus* was not found in the big batch, the genus dominated from days 5 to 14 (7.5–17.6%) in the small batch. The OTUs exhibited 99.3–100% identities with *Enterococcus gallinarum*. The genome sequences of the species encoding all EET genes found in *L. monocytogenes* are as follows: FAD transfer (*ecfA* and *A’*, *fmnA*, and *fmnB*), electron transfer (*pplA*, *ndh2*, *eetB*, and *eetA*) and DMK processing components (*dmkA* and *dmkB*).

### 2.8. KEGG Ortholog Related to Dyeing Intensity and Initiation of Indigo Reduction

As listed in Appendix A, 128 items of KEGG orthology exhibiting a correlation coefficient of ≥0.9 with the dyeing intensity were selected from 5502 KEGG orthology items in “pred metagenome unstrat” created with PICRUSt2 analysis in the big batch (Appendix A). The listed KEGG orthology included functions of “prokaryotic defense system”, “carbohydrate metabolism”, and “replication and repair” that were considered as related to indigo reduction in BURRITO analysis. Since items related to energy metabolism and NAD(P)H production frequently were observed in the listed KEGG orthology, to understand the taxa contributing to the initiation of indigo reduction, we tried to identify the taxa contributing to the items related to energy metabolism and NAD(P)H production in “CountContributedByOTU”, which created with PICRUSt2 analysis in day 2 sample (Table 2). In the case of plural subunits listed in Appendix A, one subunit among them was listed in Table 2. In all the items, contributions by *Alkalibacterium*, which is the most predominant taxon (38.9% in the microbiota) (Figure 1), were the largest (45.3–100%), and these contributing ratios were higher than the existing ratio (38.9%) in the microbiota. Especially, the contribution higher than 80% in H^+^/Na^+^-translocating ferredoxin:NAD^+^ oxidoreductase (EC:7.1.1.11 7.2.1.2) (84.2%), malate dehydrogenase (decarboxylating) (EC:1.1.1.39) (100%) and thioredoxin 2 (EC:1.81.8) (96.6%). *Alcalicella* contributed to V/A-type H^+^/Na^+^-transporting ATPase (EC:7.1.1.2 7.2.2.1) (4.4%) and ferredoxin/flavodoxin-NADP^+^ reductase (EC:1.18.1.1 1.19.1.1) (7.4%) that was lower than the existing ratio (8.1%) in the microbiota. Although the ratio in the microbiota and the contributing ratio were rather small, *Amphibacillus* (2.7% in the microbiota) contributed to V/A-type H^+^/Na^+^-transporting ATPase (2.9%), alcohol dehydrogenase, propanol-preferring (EC:1.1.1.1) (3.5%), and ferredoxin/flavodoxin-NADP^+^ reductase (2.5%).

Correlation coefficients between KEGG orthology items and dyeing intensities in the small batch were estimated, and highly correlated items were selected (Appendix A). However, the values of correlation coefficients were not as high as that in the big batch, and items that exhibited correlation coefficients of ≥0.77 were selected (131 items) from 5924 KEGG orthology items. In this case, the items that produced NADH and ATP showed a high correlation. Although the correlation coefficients were lower and the order of the items exhibiting a high correlation coefficient was different from the results of the big batch, the listed items were similar to the results of the big batch. As dyeing intensity was high on day 7 within the first 2 weeks of the fermentation, we attempted to identify the taxa contributing the items related to ATP and NAD(P)H production in “CountContributedByOTU” in the day 7 sample (Appendix A). Because *Alkalicella* was the predominant taxon in the sample (50.0%), its contributions to H^+^/Na^+^-translocating ferredoxin:NAD^+^ oxidoreductase (46.1%) and V/A-type H^+^/Na^+^-transporting ATPase (46.5%) were high. Although the existing ratio was smaller than in the big batch, *Alkalibacterium* possessed a wider range of NADH-producing enzymes than the other taxa. Considering the existing ratio of *Alkalibacterium* (9.6%), its contribution to H^+^/Na^+^-translocating ferredoxin:NAD^+^ oxidoreductase was high (39.9%). Although the contribution to H^+^/Na^+^-translocating ferredoxin:NAD^+^ oxidoreductase (6.9%) by *Enterococcus* was almost equal to the existing ratio in the microbiota (8.8%), the taxon highly contributed V/A-type H^+^/Na^+^-transporting ATPase (26.2%) considering the existing ratio. As described above, it was considered that major taxa play roles in introducing energy and activating transporters for EET in both big and small batches.

## 3. Discussion

There are many complex microbial communities consisting of many taxa that serve their objective functions (e.g., foods and beverages produced via natural fermentation, livestock feed silage, and wastewater treatment systems). In such cases, appropriate convergence of the microbiota toward objective functions or objective states is desirable because the initial state of the microbial communities should be changed to objective states. Therefore, it is important to understand the dynamics and principles of microbiota transition associated with environmental changes. In this study, we clarified how differences in the fermentation scale affect the transitional changes from the original microbiota (in *sukumo*) to the indigo-reducing state in the indigo-reducing microbial system. Big-scale fermentation produces a stable low-ORP (<−600 mV) environment rapidly (within day 2). This rapid decrease in ORP in the big batch excluded *Actinomycetota* and other unnecessary microorganisms for indigo reduction effectively and resulted in the predomination of *Alkalibacterium*, which largely and effectively contributed to indigo reduction [1,13]. This means that a slow decrease in ORP provides an adaptation opportunity for *Actinomycetota* and other microorganisms that have difficulty adapting to the rapid decrease in ORP. The survival of these microbes makes subsequent relationships between microbes more complex. In the interbacterial relationship analysis, only positively related bacterial networks were constructed in the big batch, and the extent of microbiota changes was smaller than those in the small batch. It is thought that bacterial communities causing negative relationships are eliminated by rapid ORP reductions within day 2 in the big batch. In contrast, the microbial community that could metabolize oxygen in the initially constructed network was changed to a bacterial network acclimated to the fermentation environment to indigo reduction in the small batch. The former network is negatively related to the latter. The networks of indigo-reducing bacteria comprise a few taxa derived from the initial and the later bacterial networks. The network analyses suggest that a slow ORP decrease at the beginning of the fermentation caused many taxa in *sukumo* to survive; these surviving taxa comprised the major network circles. On the other hand, a rapid ORP decrease selected a proper bacterial network adapted to the low OPR and high pH.

The origin of the indigo-reducing microbiota is *sukumo* [9]. The major constituent taxa in *sukumo* are *Actinobacteria*, *Gemmatimonadetes*, *Pseudomonadota* (formerly *Proteobacteria*), and *Bacillota* [9,13]. Because the major members of the microbiota are unrelated to indigo reduction, the convergence of the microbiota to the microbiota contributing to the indigo reduction state is not easy. As an initial step in the fermentation, *sukumo* was treated with hot wood ash extract (60–80 °C, pH > 10.5). It has been reported that high pH has a larger impact on the microbiota in *sukumo* than in a high-temperature treatment (60 °C) [10]. High temperature is effective if combined with high pH. The introduction of wheat bran into the fermentation batch was close to the direction of indigo reduction in multivariate analysis, whereas pH was in the same direction, leading the fermentation to decrease the ORP [10]. Thus, high pH and low ORP lead to changes in the original microbiota toward adaptability in alkaline anaerobic environments; wheat bran brings about the further selection of microorganisms and introduces an energy source for EET. By comparing big- and small-scale fermentation, we understand for the first time the large impact of rapid decrease in ORP on the convergence of the microbiota toward the occurrence of EET because our previous study was performed in a small batch. Based on the present and the previous results [10], a preparation of appropriately high pH (pH 10.0–11.0) and rapid ORP decrease (<−600 mV) in the big batch (≥250 L), and introduction of wheat bran within days 3–5 from the initiation of fermentation will converge the original microbiota in *sukumo* to indigo reduction rapidly.

Because *E*°’ of NADH is −601 mV at pH 10, whether NADH can be the electron source for indigo, which requires at least *E*°’= −600 mV for reduction, is controversial [3,18]. In other words, there is an opinion that there should be a supply of electrons from a substance with even lower *E*°’. For example, if electron release from acetaldehyde (NAD-independent) is possible, the reducing power is much higher than NADH, its *E*°’= −1044 mV at pH 10. However, FMN-dependent NADH-azoreductases have been reported to reduce indigo in vitro [19,20]. Although it is difficult to explain how NADH, which is present only intracellularly, transfers electrons to extracellular indigo, FMN and NADH can act as an electron mediator and donor, respectively, in vitro. It is difficult to consider that NADH reacts directly to indigo; thus, we estimated the possibility of FMN or riboflavin transferring electrons from the cell to indigo; FMN and riboflavin show *E*°’= −624 mV and −612 mV, respectively, at pH 10.5 and can thus contribute to indigo reduction (Appendix A).

The EET system found in *L*. *monocytogenes* is present in hundreds of species across the phylum *Bacillota*, including obligate anaerobic, aerotolerant, and facultative anaerobic species within the orders *Eubacteriales*, *Bacillales*, and *Lactobacillales* [16]. As the major constituents of bacteria found in indigo fermentation belong to these orders [10,13], we attempted to identify the EET-responsible gene sequences in the major constituent bacteria in indigo fermentation. The EET system consists of genes for the transportation of FAD to modify the FMN-associated protein family, the electron transfer protein family from NADH to the FMN-associated family, and the synthesis of DMK, which is an alternative quinone separated from the aerobic respiratory system [16]. Although we could identify these series of genes in major constituent bacteria in big and small batches, the constituent genes and their locations were different from *L*. *monocytogenes*. This is probably because there are many variations in EET gene configuration among members of the phylum *Bacillota*. Variations in the configuration of EET genes have been reported [16,21,22].

The KEGG orthologs that were highly related (correlation coefficient of ≥0.9) to dyeing intensity in the big batch did not indicate a direct correlation with EET-related proteins. However, numerous KEGG orthologs related to the production of ATP and NAD(P)H, which are related to EET, have been identified. ATP is necessary for the transport of substrates and electron mediators across the membrane for EET energization and construction. The meaning of various NAD(P)H production systems is that many NAD(P)H synthetic pathways support electron transfer in EET systems. Thus, it is thought that the extent of the expansion of the EET system should be limited, considering the bacterial cell structure and total metabolic system of each microorganism. However, a larger enhancement of energization is necessary for material transportation and energy supply required for the invigoration of EET compared to the expression of proteins in the EET system.

The rapid decrease in ORP produced in the big batch excluded *Actinomycetota* effectively and *Alkalibacterium*, which contributed largely to indigo reduction. This means that *Alkalibacterium* can be predominated in the microbiota if we can prepare a condition with rapid ORP decrease (≤−600 mV) using appropriate raw materials, especially *sukumo*. This indicates that the functional superiority of *Alkalibacterium* for being predominant in the microbiota was greater than that of the other taxa. This may be due to the functional superiority of *Alkalibacterium* in “prokaryotic defense system”, “starch and sucrose metabolism”, and “replication, recombination, and repair proteins”. These functions are related to the dyeing intensity. In *L*. *monocytogenes*, when regeneration of NAD^+^ is properly performed by aerobic respiration and EET, cells maintain their integrity and pathogenicity [23]. However, if regeneration of NAD^+^ does not appropriately proceed, cell lysis occurs, and pathogenicity is lost. Referring to this fact, the functions of “prokaryotic defense system”, “starch and sucrose metabolism”, and “replication, recombination, and repair proteins” may be related to each other to maintain the integrity of bacterial cell metabolism in the corresponding ecosystem. Nutrients derived from *sukumo* are consumed during the first month of fermentation. During this period, “prokaryotic defense system”, “starch and sucrose metabolism”, and “replication, recombination, and repair proteins” functions are developed. However, after the first month of fermentation, substrate circulation occurred within the fermentation system, with wheat bran added after day 16 as the only externally introduced substrate. During this period, the EET activity gradually decreased with aging. However, this state is very stable, and the microbiota change is very slow.

This study has some limitations. This study employed a fermentation fluid used in a craft center obtained by transportation from the craft center. Therefore, the sample condition, including the microbiota of day 212, which exhibited weak indigo reduction, may have been changed during the transportation. In addition, although we attempted to make the laboratory fermentation resemble big-scale fermentation as much as possible, it was difficult to use the same preparation and maintenance technique of the fermentation fluid as well as the frequency of dyeing.

## 4. Materials and Methods

### 4.1. Preparation of the Big-Scale Batch in a Craft Center

A big-scale (270 L) batch of *sukumo* fermentation was prepared in a craft center in Miura, Kanagawa, Japan (35°09′19.6″ N 139°36′53.3″ E). On day 1, 28.1 kg of *sukumo*, which was produced by O.N. in Tokushima, Shikoku, Japan, was mixed with 40 °C wood ash extract to obtain a clay-like state on a vat and kept at 30 °C. Wood ash extract was made by mixing wood ash produced by burning oak trees with hot water (100 °C) at a ratio of 1:20–40. The following day (day 2), the treated *sukumo* was transferred to a big jar, and 160 L (60% of full volume; 40 °C) of wood ash extract was mixed. Wheat bran (300 g) was added when the fluid temperature was lowered to 30 °C. On the next day (day 3), the rest of the 40% volume of wood ash extract (40 °C) was introduced. The surface area of the fermentation liquid was approximately 0.37 m^2^. The surface-area-to-volume ratio was approximately 1:722. Shell lime (Eco-organic House, Fukuoka, Japan) was occasionally added to the fermentation fluid to maintain a pH ≥ 10.3. Wheat bran was added to feed the microbiota in the fermentation fluid on days 20 (125 g), 73 (140 g), 108 (140 g), and 186 (125 g). The fermentation fluid was maintained by checking the pH and dyeing intensity on-site by a craftsperson. Although fermentation was maintained for more than one year, samples for analysis were taken on days 3 to 212 (days 3, 10, 27, and 212). The obtained aliquot of duplicate samples was transferred to the laboratory in Sapporo (43°01′12.0″ N 141°25′01.4″ E) by mail for sample preparation for next-generation sequencing.

### 4.2. Preparation of the Small-Scale Batch in the Laboratory

Similar to the procedure, including the used materials prepared for the big-scale batch in the craft center, we prepared a small-scale batch (5 L) of *sukumo* fermentation under laboratory conditions. On day 1, 520 g of *sukumo* produced by O.N. was mixed with 40 °C wood ash extract to make a clay-like state on a vat and kept at 30 °C. The wood ash extract was prepared by boiling tap water (5 L) for 10 min with 125 g of wood ash (Nagomi Co., Gobo, Wakayama, Japan) produced by burning *Quercus phillyraeoides* A. Gray. The following day (day 2), the *sukumo* was transferred to a small jar, and 3 L (60% of full volume; 40 °C) of wood ash extract was mixed with the *sukumo*. Wheat bran (2 g) was introduced when the fluid temperature lowered to 30 °C. On day 3, the rest of the 40% volume of wood ash extract was added to the batch. The surface area of the fermentation liquid in the jar was approximately 0.053 m^2^. The surface-area-to-volume ratio was 1:94. Shell lime (Eco-organic house) was occasionally added, as described above. Wheat bran (5 g) was added to feed the microbiota in the fermentation fluid on days 13, 44, 84, and 96. The samples were taken on days 2, 5, 6, 7, 14, 29, 96, and 200.

### 4.3. Measurements of Fermentation Fluid

The pH and ORP of the fermentation fluid were measured using a D-71 pH meter (Horiba, Kyoto, Japan) and a D-75 pH/ORP/DO meter (Horiba), respectively. The reducing state of indigo was determined based on the dyeing intensity of the fermentation fluid. After dipping a small piece of cotton cloth in the fermentation fluid for 1 min, it was exposed to air. After 2–5 min of exposure, the cloth was washed with tap water and dried. The pH, ORP, and dyeing intensity of the big batch were estimated when the samples from the big batch arrived at the laboratory. Dyeing intensity was analyzed using Mathematica (version 12.2). The intensity was expressed as *L***a***b** value (defined by the Commission Internationale de l’éclairage (CIE)), which is the square root of *L**^2^ + *a**^2^ + *b**^2^. *L***a***b** color space is represented by *L** and *a** + *b**, representing lightness and chromaticity, respectively. The extensions of *a** and *b** represent from red (*a**) to green (−*a**) and from yellow (*b**) to blue (−*b**). The staining intensity was expressed as *L***a***b** value × 10.

### 4.4. Illumina MiSeq Sequencing

DNA was extracted using the FastDNA Spin kit for soil (MP Biomedicals, Santa Ana, CA, USA), according to the manufacturer’s instructions. The bacterial 16S rRNA gene sequence of the V3–V4 region was amplified using the primer pair associated with the overhang for 2nd PCR, random sequences of 0–5 base(s) (described as N; adapter for quality control) and a gene sequence for the amplification of partial 16S rRNA (341F and 805R): V3V4f_MIX (ACACTCTTTCCCTACACGACGCTCTTCCGATCT-NNNNN-CCTACGGGNGGCWGCAG) and V3V4r_MIX (GTGACTGGAGTTCAGACGTGTGCTCTTCCGATCT-NNNNN-GACTACHVGGGTATCTAATCC). The 1st PCR solution (10 μL) consisted of 5 μL of 2× PCR buffer of KOD FX Neo (TOYOBO, Osaka, Japan), 2 μL of dNTPs (each 2 mM; TOYOBO), 0.2 μL of each of the forward and reverse primers, 1 μL of template DNA, and 0.2 μL of 1 U/mL KOD FX Neo polymerase (TOYOBO). The cycling conditions were as follows: 94 °C for 2 min; 25–35 cycles of 94 °C for 10 s, 55 °C for 30 s and 68 °C for 30 s; extension of 68 °C for 7 min. The amplified products were purified using AMPure XP beads (Beckman Coulter Genetics, Danvers, MA, USA). The second PCR was performed with index-adapted primers as follows: 94 °C for 2 min; 8–12 cycles of 94 °C for 30 s, 60 °C for 30 s, and 72 °C for 30 s; and extension of 72 °C for 5 min. The amplified library was also purified using AMPure XP beads (Beckman Coulter Genetics) for NSG using the MiSeq platform (Illumina, San Diego, CA, USA) and MiSeq reagent kit v3 (Illumina).

### 4.5. Sequencing Data Analysis

Because the sequences of 1st primers and N-sequences remained in the resulting sequences from NGS, they were removed from the sequences using Cutadapt version 1.18. Thus, the obtained fastq files were processed using QIIME2 ver. 2020.2 [24]. Paired-end read merging, quality control, and amplicon sequence variants (ASVs) were created using DADA2 [25]. Taxonomic identification of the obtained sequences was performed using a feature classifier that specified the primer pair 341F–805R based on the Silva database [26,27]. For an updated taxonomic identification, the representative sequences of the major members in each sample were used for a BLAST search at the National Center of Biotechnology Information (NCBI; https://blast.ncbi.nlm.nih.gov/Blast.cgi, accessed on 24 May 2023). Rarefaction curves of the observed operational taxonomic units (OTUs) and Shannon index of diversity [28] were estimated using the QIIME2 alpha diversity script. Linear discriminant analysis (LDA) effect size (LEfSe) of the identification data based on the 16S rRNA gene sequence was conducted using the Galaxy site of the Huttenhower Lab (https://huttenhower.sph.harvard.edu/galaxy, accessed on 21 August 2023). Relationships between the big- and small-scale batches microbiota were estimated by redundancy analysis (RDA), which was performed using the R package (vegan ver. 2.6-4, ggrepel ver. 0.9.2, ggplot2 ver. 3.4.0, ggpubr ver. 0.5.0). Spearman’s rank correlations between the genera in changes of the relative content with fermentation aging were calculated using R (programs Psych and Reshape2). Cytoscape software version 3.9.1 was used to visualize the interactions among the genera. The predictive function of the metagenome in the abundance of the Kyoto Encyclopedia of Genes and Genomes (KEGG) orthology was estimated using PICRUSt2 [29]. Based on the result of PICRUSt2 (pred_metagenome_contrib.legacy) combined with the taxonomic data from QIIME2 analysis, taxa-functional relationships were analyzed using BURRITO (http://elbo.gs.washington.edu/software_burrito.html, accessed on 28 March 2023) [30]. To understand the functional contribution of dying intensity, the correlation between changes in dyeing intensity depending on fermentation aging and the intensity of centered logratio transformed values of the KEGG ortholog (pred_metagenome_unstrat) from PICRUSt2 analysis was estimated. The genes series of EET, which was found in *L*. *monocytogenes* [18], was annotated using the DDBJ Fast Annotation and Submission Tool (DFAST) pipeline (https://dfast.nig.ac.jp/, accessed on 20 April 2023) in the genome whose taxa (≥99.0% identities) involve in the indigo fermentation bath. Genome sequences of taxa related to the major constituents of indigo fermentation fluid were obtained from NCBI (https://www.ncbi.nlm.nih.gov/, accessed on 12 April 2023).

### 4.6. E°’ Estimation of Electron Mediator Candidates for Possible Electron Donor for Indigo

To estimate the possibility of flavins for electron donors as indigo midpoint, redox potentials of the candidate substance at a high pH were estimated using Equation (1).
*E*°’_pH2_ = *E*°’_pH1_ − 2.303 *m*RT/*n*F(pH_2_ − pH_1_)(1)
where *m* and *n* are the numbers of protons and electrons related to the reduction, respectively, R = gas constant (8.315 J·K^−1^·mol^−1^), T = absolute temperature (298 K = 25 °C), and F = Faraday constant (96.486 kJ·mol^−1^·V^−1^), −2.303 RT/F = approximately 59 mV at 25 °C. The values “V vs. the standard hydrogen electrode (SHE)” were used from Ref [31].

## 5. Conclusions

The microbiota responsible for indigo reduction originates from autochthonous microorganisms in *sukumo* (composted leaves of *Polygonum tinctorium* L.). However, most autochthonous microorganisms in *sukumo* are not directly associated with indigo reduction. Therefore, the microbiota in *sukumo* must converge appropriately to reduce indigo. The methodology for the preparation of *sukumo* fermentation was established based on the experience of the craftspersons. However, the relationship between the environmental parameters and changes in the microbiota from the original state of *sukumo* has not been thoroughly understood, and accordingly, the convergence of microbiota in different fermentation scales was analyzed. The rapid ORP decrease in the big batch excluded aerobic bacteria, mainly *Actinomycetota,* effectively and increased aerotolerant *Alkalibacterium*, which can reduce indigo. It is possible that indigo reduction occurs via mechanisms similar to that of EET in *L*. *monocytogenes*. Gene sequences related to EET were identified in major species in the microbiota of the indigo fermentation fluid. The correlation between the intensity of indigo reduction and metabolic functions in the KEGG orthologs suggested that V/A-type H^+^/Na^+^-transporting ATPase and NAD(P)H-producing enzymes drive membrane transportation of electron mediators and substrates and energize EET system, respectively. Based on these results, it can be concluded that EET occurs from intracellular NAD(P)H to extracellular free flavin, and extracellular FMNylated protein originates from the anaerobic electron transfer system.

## Figures and Tables

**Figure 1 ijms-24-14696-f001:**
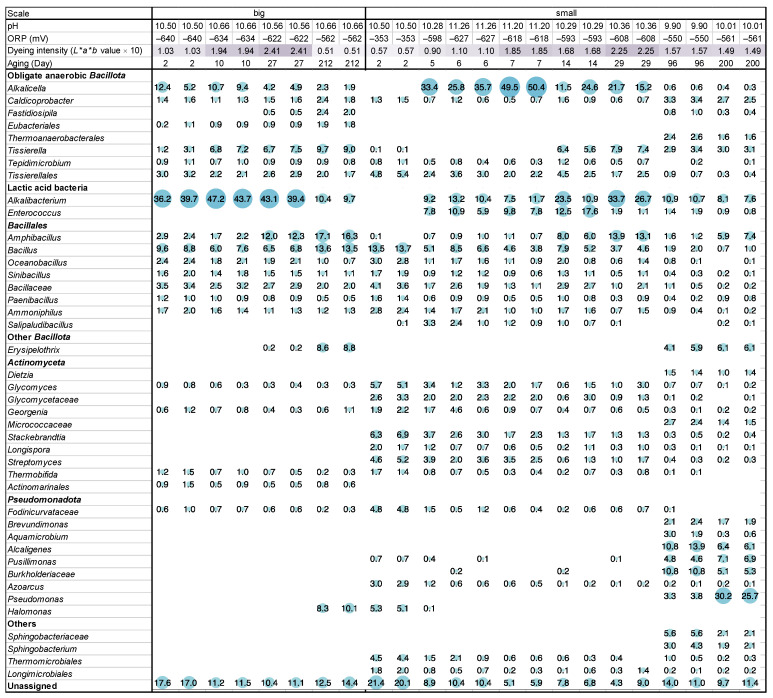
Changes in the relative abundance of bacterial communities (≥2% in any sample), pH and redox potential (ORP), and dyeing intensity depend on the fermentation age in indigo fermentation fluid big and small batches based on 16S rRNA analysis. Circle sizes correspond to abundance, as shown as percentages.

**Figure 2 ijms-24-14696-f002:**
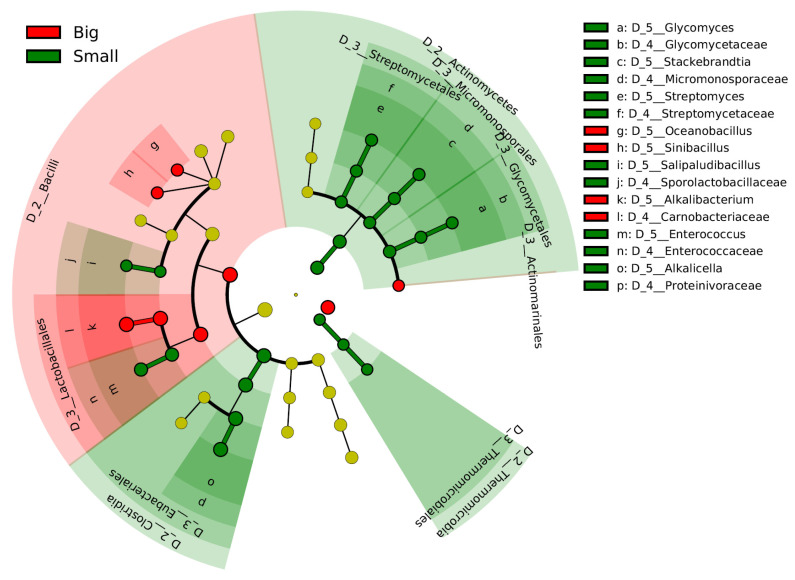
Bacterial markers of big and small batches in the early phase of fermentation. Days 2–27 in the big batch were compared with days 6–29 in the small batch. The linear discriminant analysis (LDA) effect size (LEfSe) analysis was performed to identify the markers for each group (significant when LAD score > 3.6).

**Figure 3 ijms-24-14696-f003:**
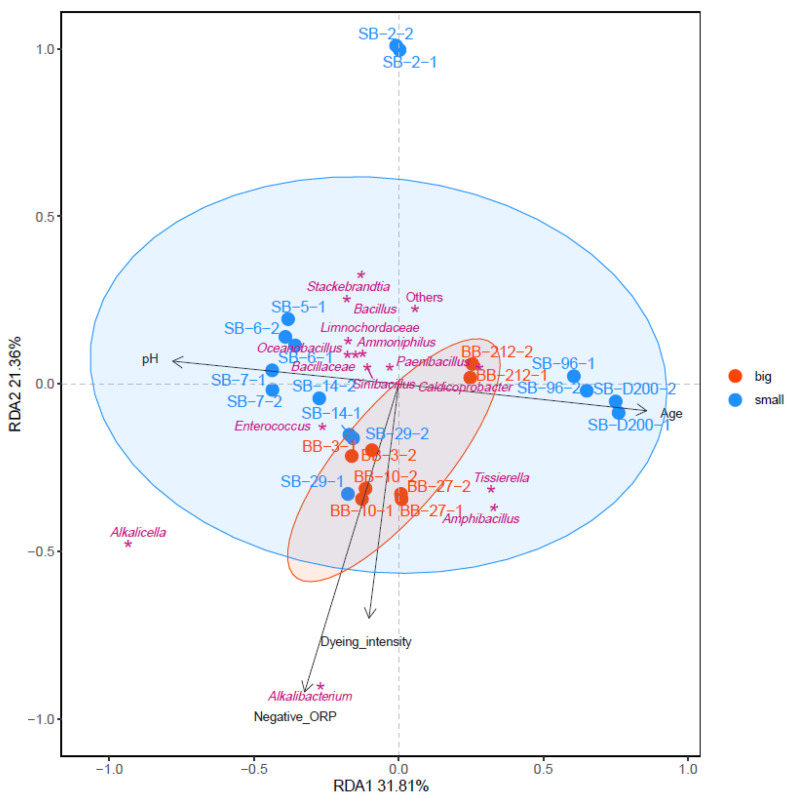
Redundancy analysis (RDA) of the microbial community in big (red-filled circles)- and small (blue-filled circles)-scale indigo fermentations. Purple asterisks represent the core microbiota. Black arrows indicate different environmental factors.

**Figure 4 ijms-24-14696-f004:**
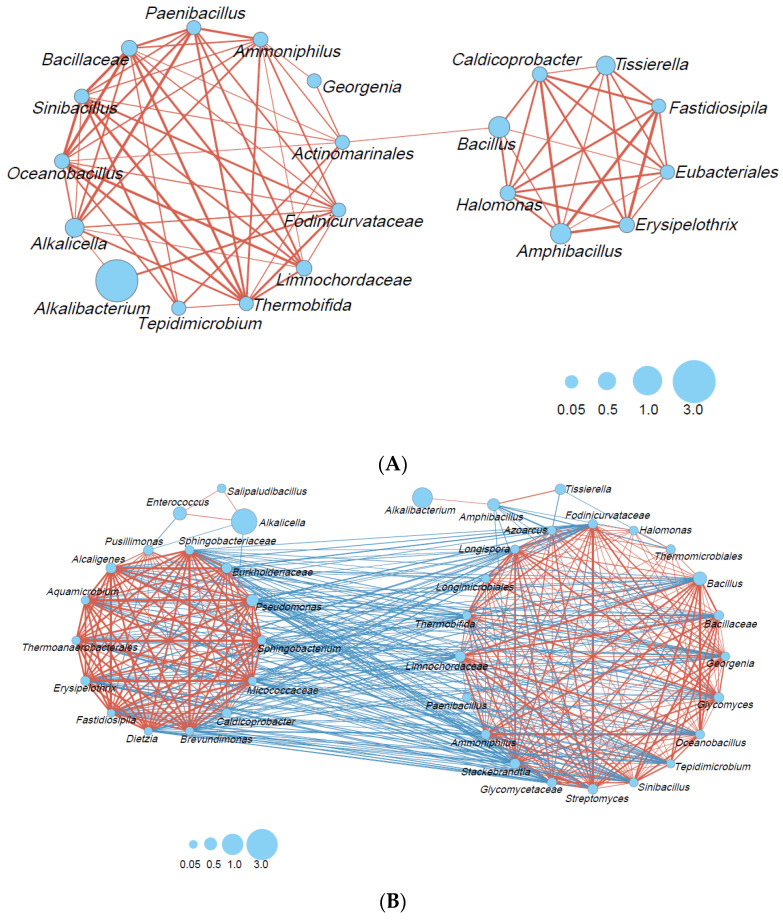
Bacterial community networks based on the relative content change trend analyzed with Spearman’s rank correlation coefficient (rs > 0.6; *p* < 0.05): (**A**) big-scale batch, days 3–212; (**B**) small-scale batch, days 2–200. The red and blue lines represent positive and negative correlations, respectively. The thickness of the lines corresponds to the strength of the relationship (0.6 < rs ≤ 1), whereas the circle size shows accumulated taxon abundance during each fermentation period.

**Figure 5 ijms-24-14696-f005:**
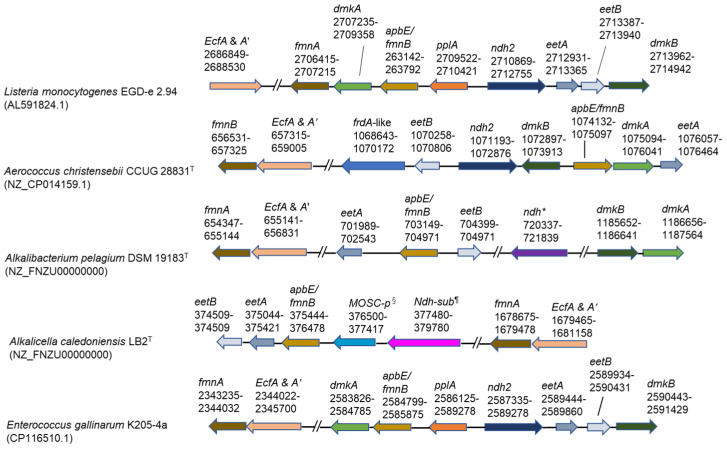
Genetic loci associated with extracellular electron transfer genes. Top two rows of the panel show reported results [16,17]. The other three indicate predominant taxa found in this study. The frdA-like gene was reported by Light et al. [17]. *: NADH dehydrogenase, which is not similar to ndh-2; ^§^: MOSC domain-containing protein; ^¶^: NADH dehydrogenase subunit, which is not similar to ndh-2. Superscripts of “T” attached to the strain names indicate type strain.

**Table 1 ijms-24-14696-t001:** Functional contributions of constituents of taxa in big-scale batch on day 27. Important subpathways for indigo reduction were selected based on their relationship with dyeing intensity. *: constituted ratio in the microbiota; ^§^: the functional abundance ratio in the whole subpathway of the sample.

**Category/**Superpathway/Subpathway	*Alkalibacterium*	*Amphibacillus*	*Bacillus*	*Tissierella*	*Alcalicella*
	(41.3%) *	(12.2%)	(6.6%)	(7.1%)	(4.6%)
**Cellular Processes**					
Transport and catabolism					
Prokaryotic Defense System (1.57%) ^§^	64.8%	9.1%	4.8%	6.3%	2.6%
**Metabolism**					
Carbohydrate metabolism					
Starch and sucrose metabolism (1.84%)	64.9%	14.0%	5.4%	2.2%	1.2%
**Unclassified**					
Genetic information processing					
Replication, recombination, and repair proteins (1.99%)	65.0%	8.3%	5.8%	5.3%	2.1%

**Table 2 ijms-24-14696-t002:** Functional contribution for The KEGG orthologies related to NADH production or ATP synthesis by predominated constituent taxa at day 2 in the big batch. * Correlation coefficient between the fluctuation of KEGG orthology and dyeing intensity. ^§^ Existing percentage among the microbiota.

KEGG Orthology/Correlation Coefficient	Description of KEGG Orthology/Reaction/Contributed Taxa in the Microbiota	Functional Contribution (%) by OTUs Involving Taxon
K02117 (0.95) *	V/A-type H^+^/Na^+^-transporting ATPase subunit A (EC:7.1.2.2 7.2.2.1) ADP + Pi ↔ ATP	
	*Alkalibacterium* (38.9%) ^§^	71.3%
	*Alkalicella* (8.1%)	4.4%
	*Amphibacillus* (2.7%)	2.9%
K03615 (0.93)	H^+^/Na^+^-translocating ferredoxin:NAD^+^ oxidoreductase subunit C (EC:7.1.1.11 7.2.1.2) Ferredoxin (red) + NAD^+^ ↔ ferredoxin (ox) + NADH + H^+^	
	*Alkalibacterium* (38.9%)	84.2%
	*Amphibacillus* (2.7%)	1.7%
K13953 (0.90)	Alcohol dehydrogenase, propanol-preferring (EC:1.1.1.1) Primary alcohol + NAD^+^ ↔ Aldehyde + NADH + H^+^	
	*Alkalibacterium* (38.9%)	65.4%
	*Bacillus* (9.2%)	4.5%
	*Amphibacillus* (2.7%)	3.5%
	*Georgenia* (1.0%)	2.6%
	*Tissierellales* (3.1%)	2.1%
	*Glycomyces* (0.9%)	2.0%
K00028 (0.91)	Malate dehydrogenase (decarboxylating) (EC:1.1.1.39) (S)-Malate + NAD^+^ ↔ Pyruvate + CO_2_ + NADH + H^+^	
	*Alkalibacterium* (38.9%)	100%
K03672 (0.91)	Thioredoxin 2 (EC:1.81.8) Protein dithiol + NAD^+^ ↔ Protein disulfide + NADH + H^+^	
	*Alkalibacterium* (38.9%)	96.6%
K00528 (0.87)	Ferredoxin/flavodoxin-NAD(P)^+^ reductase (EC:1.18.1.2 1.19.1.1) Ferredoxin/flavodoxin (red) + NAD(P)^+^ ↔ Ferredoxin/flavodoxin (ox) + NAD(P)H + H^+^	
	*Alkalibacterium* (38.9%)	45.3%
	*Alkalicella* (8.1%)	7.4%
	*Amphibacillus* (2.7%)	2.5%
	*Sinibacillus* (1.8%)	2.4%

## Data Availability

The datasets of NGS results in this study can be available in online repositories. The names of the repository and accession number can be found at: http:www.ddbj.nig.ac.jp/, DRA016199, accessed on 27 September 2023.

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
