# Peer review of "Effect of Fermentation Scale on Microbiota Dynamics and Metabolic Functions for Indigo Reduction"

_ijms, 2023, doi:10.3390/ijms241914696_

Round 1

Reviewer 1 Report (New Reviewer)

The article "Effect of Fermentation Scale on Microbiota Dynamics and Metabolic Functions for Indigo Reduction" explores the convergence mechanisms of microbiota and microbial physiological basis for indigo reduction, analyzing microbiota led by different velocities in redox potential (ORP) decrease at different fermentation scales. The study found that appropriate convergence of the microbiota towards objective functions or objective states is desirable for indigo reduction during fermentation. The rapid decrease in ORP in the big batch excluded Actinomycetota and other unnecessary microorganisms for indigo reduction effectively and predominated Alkalibacterium, which largely contributed to indigo reduction effectively. The article provides insights into the dynamics and principles of microbiota transition associated with environmental changes during indigo dyeing fermentation.

Please revise the manuscript by a native English speaker

Some corrections:

line 21-22 – please correct “contributed to 21 largely to the effective” – revise the whole manuscript

Line 306 – Please correct the scientific names to italics – revise for similar errors

Line 498 – Please specify the origin of wheat bran

Please specify the limitations of the study

After the implementation of these corrections, the article can be considered for publication

Please revise the manuscript by a native English speaker

Author Response

Comments for revivers

Reviewer 1

(x) Minor editing of English language required

Does the introduction provide sufficient background and include all relevant references?

( )           (x)          ( )           ( )

Are the results clearly presented?

( )           (x)          ( )           ( )

→ We tried to improve the descriptions of “Introduction” and “Result”.

Comments and Suggestions for Authors

Please revise the manuscript by a native English speaker

→ The manuscript has been checked by native English speakers before submission of this manuscript. However, there are several parts to revise in the manuscript. Therefore, the manuscript has been checked by native English speakers again.

Some corrections:

line 21-22 – please correct “contributed to 21 largely to the effective” – revise the whole manuscript

→ Thank you for your comment. We revised as "largely contributed to the effective---".

Line 306 – Please correct the scientific names to italics – revise for similar errors

→ If I misunderstood your comment, I am very sorry. There is no scientific name to italics.

Line 498 – Please specify the origin of wheat bran.

→ We specified the time the wheat bran added.

Please specify the limitations of the study

→ We described the limitations of this study at the end of “Discussion”.

Reviewer 2 Report (New Reviewer)

The authors conducted an interesting study related to the decomposition of indigo-producing plants, in the case of microbiota change along with the genomic analysis. The article itself is interesting, and worthy of publication if the following issues can be addressed:

a. Redraw table 1 as some of the materials seem to be truncated;

b. The font of session 4.5 is not consistent with other part of manuscript. please be consistent.

c. Uniform all the references.

Author Response

Reviewer 2

a. Redraw table 1 as some of the materials seem to be truncated;

→ Thank you for your comment. We revised. We hope it became better.

b. The font of session 4.5 is not consistent with other part of manuscript. please be consistent.

→ The font of section 4.5 has been unified to Palatino Linotype and the font size has been unified to 10.

c. Uniform all the references.

→ We unified the format of all the references.

This manuscript is a resubmission of an earlier submission. The following is a list of the peer review reports and author responses from that submission.

Round 1

Reviewer 1 Report

1.     English should be improved throughout the manuscript.

 2. Quantitative information should be provided in the abstract.

3. The concussion should be concise and to the point indicating the application of the work.

 4. The novelty of the work should be established.

5. Please write one paragraph in the introduction about the water pollution problem,

6.add the material and methods part

7.Please write one paragraph in the introduction onto advanced materials, in general,

8. Please provide error graphs in the figure; where are required.

9. Please improve the quality of the Figures. 

10.Please compare your results with previous studies and mention clearly how your work is important in comparison to already been reported.

OK